

# Application of unsupervised pattern recognition approaches for exploration of rare earth elements in Se-Chahun iron ore, central Iran.

Mohammadali Sarparandeh [1], Ardeshir Hezarkhani [1]

[1]Department of Mining and Metallurgical Engineering, Amirkabir University of Technology, Tehran, +9821, Iran

*Correspondence to*: Mohammadali Sarparandeh (masarparandeh@aut.ac.ir)

**Abstract.** The use of efficient methods for data processing has always been of interest by researchers in the field of earth science. Pattern recognition techniques are appropriate methods for high-dimensional data such as geochemical data. Evaluation of geochemical distribution of REEs needs to use such methods. Especially multivariate nature of REEs data

makes it a good target for numerical analysis. The main subject of this paper is application of unsupervised pattern recognition approaches in evaluating geochemical distribution of rare earth elements (REEs) in the Kiruna type magnetite–apatite deposit of Se-Chahun. For this purpose, 42 bulk lithology samples were collected from Se-Chahun iron ore deposit. In this study, 14 rare earth elements were measured with ICP-MS. Pattern recognition makes it possible to evaluate the relations between the samples based on all these 14 features, simultaneously. In addition to providing easy solutions,

discovery of the hidden information and relations of data samples is the advantage of these methods. Therefore, four clustering methods (unsupervised pattern recognition) including modified basic sequential algorithmic scheme (MBSAS), hierarchical (agglomerative), k-means and self-organizing map (SOM) were applied and results were evaluated using silhouette criterion. Samples were clustered in four types. Finally, the results of this study were validated with geological facts, and analysis results such as SEM, XRD, ICP-MS and optical mineralogy. The results of k-means and SOM have the

best matches with reality, experimental studies of samples and also field surveys. Since only the rare earth elements are used in this division, a good agreement of the results with lithology is considerable. It concluded that the combination of the proposed methods and geological studies, leads to finding some hidden information and this approach has the best results compared to using only one of them.

**Key words:** geochemical exploration of REEs, unsupervised pattern recognition, geochemistry of Se-Chahun, central Iran.

## 1.   Introduction

In present study, geochemical distribution of REEs is evaluated using bulk lithology samples. Clustering approach attempts to organize unlabeled feature vectors into clusters (natural groups) such that samples within a cluster are similar to each other but differ from those in other clusters (Hilario and Ivan, 2004). Clustering analysis is an important and useful tool for analyzing large datasets that contain many variables and experimental parameters. Therefore, the application of cluster





analysis to complex datasets has attracted a high level of scientific interest in various aspects of geochemistry researches (Nguyen et al., 2015). In order to investigate the distribution of elements, it is essential for a robust classification scheme to cluster chemistry samples into homogeneous groups (Guler et al., 2002). Several common clustering techniques have been utilized to divide geochemical samples into similar homogeneous groups with the ultimate objective of characterizing the

quality of elements such as principal component analysis, fuzzy k-means clustering technique and Q-mode hierarchical cluster analysis to assess the chemistry of groundwater and to identify the geological factors. For example, Ji et al. (2007) developed semi-hierarchical correspondence cluster analysis and showed its application for division of geological units with the help of geochemical data that is systematically collected from an area around Tahe in Heilongjiang Province, north China. Meshkani et al. (2011) used hierarchical and k-means clustering for identifying distribution of lead and zinc in

Sanandaj-Sirjan metalogenic zone in Iran. Ziaii et al. (2009) introduced the Neuro-fuzzy method for separating anomalies and showed that this method is more efficient than using multivariate statistics. Ellefsen and Smith (2016) evaluated a clustering method called Bayesian finite mixture modeling procedure by applying it to geochemical data collected in the State of Colorado, United States of America.

The proposed method of the self-organizing maps (SOM) is likely to become a complementary or an alternative tool to the

clustering methods (Kalteh et al., 2008; Iseri et al., 2009). The SOM is related to adaptive k-means, but performs a topological feature map that is more complex than just cluster analysis. After training, the input vectors are spatially ordered in the array, i.e. the neighboring input vectors in the map are more similar than the more remote ones (Du and Swamy, 2006). The self-organizing map approach is based on unsupervised learning algorithm, and has excellent visualization capabilities including techniques that apply the reference vectors of the SOM to give an informative picture of the data (Lu et

al., 2003). Sun et al. (2009) applied SOM method to classify Pb-Zn-Mo-Ag anomalies in the mining area around Sheduolong in Qinghai Province, China. In 2012, Abedi et al. used SOM method and fuzzy k-means (FCM) to provide deposit exploration map for Now Chun copper deposit in Iran. Sarparandeh and Hezarkhani (2016) examined the application of SOM in evaluation of geochemical distribution of REEs in Choghart Fe-REE deposit in Bafq district, and showed its good performance. Generally, in cases where there are too many number of parameters and samples, pattern recognition is a

suitable approach for data processing. Exploration of rare earth elements is one of these cases because of multi elemental nature of the data. For instance, in this study, 14 rare earth elements were measured with ICP-MS. Pattern recognition makes it possible to evaluate the relations between the samples based on all these 14 features simultaneously. In addition to providing easy solutions, discovery of the hidden information and relations of data samples is the advantage of these methods.

## 2. Geological settings of study area

There are several deposit of iron ore in central and north east of Iran and Magnetite is the main mineral in most of them. In most iron ore deposits of Iran, metasomatism is the main reason of concentrating (NISCO, 1975). Systematic exploration

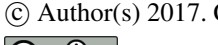



work during the 1960s and 1970s outlined 34 zones of aeromagnetic anomalies in between Bafq in the south to Saghand in the north with a total reserve of more than 1500 Mt iron ore (Torab, 2008). The Se–Chahun deposit is composed of two major groups of ore bodies called the X and XI anomalies (NISCO, 1975). Anomaly X crops out as some small black hills containing 11 Mt iron ore reserve with mainly rich magnetite ore (Torab, 2008). Anomaly XI occurs 3 km northeast of

5  anomaly X. Each anomaly consists of two or three smaller tabular to lens shaped ore bodies in association with other small bodies (Bonyadi, 2011). The mineralization is mainly hosted by metasomatized tuffs of andesite composition. Geological map of Se-Chahun deposit (anomaly X), as well as the location of samples within the study area are shown in Figure 1.

**Figure 1 Geological map of Se-Chahun deposit (anomaly X) and sample locations. Contours of open pits are shown in the map and the open pits are numbered from 1 to 4 (supplementary part of pit 2 are known as pit 4) (modified after NISCO, 1975).**

## 3. Mineralogy

The host rocks have a gradual boundary. Samples mainly include iron ores, low-grade ores (transition zone, consisting of plagioclase and actinolite) and metasomatitic rocks (mainly consisting of actinolite and plagioclase. Ore body is comprised of high grade magnetite. The most important REE bearing minerals in Se-Chahun deposit are apatite and monazite. There are two types of apatite: REE bearing apatite and depleted. Bonyadi et al. (2011) showed that some apatites of Se-Chahun have

15 been leached of LREE, Y, Na, Cl, Mg, Mn and Fe. REE bearing apatites are bright in BSE images, while leached apatites are dark. In terms of dimensions, there are two types of apatite: Coarse grain and fine grain. They can be seen under optical and scanning electron microscope. However, all of them are extremely altered and their crystals can't be seen in hand samples. The content of rare earth elements is directly related to the amount of apatite. The more the apatite, the more REE is. Monazites are very fine grains and can only be distinguished in SEM images (Figure 2). They are brighter than apatites and

20 magnetites and containing greater amounts of REEs. However, there are low amounts of monazite in samples. Therefore, apatite is the main source of REEs in Se-Chahun deposit.

## 4. Scanning electron microscopy (SEM)

Several samples were analyzed with scanning electron microscopy (SEM) and the results were used for evaluation of mineralogy and also validity of this study. Figure 2 shows the back-scattered electron images of a sample from phosphate

rocks. Monazites are brightly colored and include Ce, La and Nd. Apatites are dark gray, and include P and Ca and La but no Ce. As it can be seen in Figure 2, there are low amounts of monazite. Monazites can be seen in two ways: 1) small crystals around the apatite and 2) inclusions in apatite crystals (Figure 2, a).

**Figure 2 Back-scattered electron images of a sample from phosphate rocks. Abbreviations: Ap= apatite, Mnz= monazite and Mag= magnetite.**



## 5. Chemical analysis

In this study, 42 bulk lithology samples were collected from anomaly X of Se-Chahun iron ore deposits. They are from pit 1, 2 and 4 (supplementary part of pit 2 is known as pit 4, Figure 1). 19 samples were taken from pit 1, 9 samples from pit 2 and 14 samples from pit 4. Samples were taken from ore body and metasomatic zones. After preparation of the samples, they were analyzed with ICP-MS. The concentrations of REEs were normalized between 0 and 1 and were used as input data for clustering. These data can be divided roughly in to three groups: samples with high, medium and low concentration of REEs. Accurate determination of groups needs multivariate analysis and data processing. Another important point is that the samples are enriched by light rare earth elements (LREEs) and also Y. High amounts of REEs occurs in phosphorus iron ores and they are more in supplementary part of pit2 (or pit 4). Assayed REEs are 14 elements including La, Ce, Pr, Nd, Sm, Eu, Gd, Tb, Dy, Er, Tm, Yb, Lu and Y. mean, variance, minimum and maximum of these rare earths are presented in Table 1.

**Table 1 mean, variance, minimum and maximum of 14 assayed rare earths in 42 samples.**

## 6. Methodology

Four methods including modified basic sequential algorithmic scheme (MBSAS), hierarchical (agglomerative), k-means and SOM were applied in this study. These methods have been applied in diverse aspects of science and engineering, somewhat in geochemistry and never for exploration of REEs. The papers of Sarparandeh and Hezarkhani (2016) and Zaremotlagh and Hezarkhani (2016) are the only efforts which have been made in this area. However, there is no study that applies and compares several types of algorithms. In this study, in addition to provide such a useful information and experience, the authors show that some extra information such as the relation between REEs content and lithology of samples and can be achieved by proposed methods. Moreover, a good discrimination based on lithology attains just by using REEs. The general concepts of each method is explained in the following.

### 6.1 Sequential clustering

Sequential methods are easy and fast algorithms. These include basic sequential algorithmic scheme (BSAS) and modified BSAS (MBSAS). In BSAS two parameters should be defined by the user: the maximum number of clusters and dissimilarity threshold. The basic idea behind BSAS is that each input vector x is assigned to an already created cluster or a new one is formed. Therefore, a decision for vector x is reached prior to the final cluster formation, which is determined after all vectors have been presented. The refinement of BSAS, which is called modified BSAS (MBSAS), overcomes this drawback. The algorithmic scheme consists of two phases. The first phase involves the determination of the clusters, via the assignment of some of the vectors of X to them. During the second phase, the unassigned vectors are presented for a second time to the algorithm and are assigned to the appropriate cluster (Theodoridis & Koutroumbos, 2003). Therefore, in this study MBSAS



algorithm was applied for clustering of samples based on REEs. In this study, mean of each group and the Euclidean distance were used as the cluster centers and a measure of dissimilarity, respectively.

## 6.2 Hierarchical clustering

Hierarchical clustering procedures are among the most commonly used methods of summarizing data structure. They use a hierarchical tree which is a nested set of partitions represented by a tree diagram or dendrogram (Figure 3 and Figure 4). To separate each branch of the dendrogram, a numerical value that indicates the dissimilarity between clusters should be measured. There are several different algorithms for finding a hierarchical tree. An agglomerative algorithm begins with n subclusters, each containing a single data point, and at each stage merges the two most similar groups to form a new cluster, thus reducing the number of clusters by one. The algorithm proceeds until all the data fall within a single cluster. A divisive algorithm operates by successively splitting groups, beginning with a single group and continuing until there are n groups, each of a single individual. Generally, divisive algorithms are computationally inefficient, except where most of the variables are binary attribute variables (Webb, 2002). In this study, agglomerative approach was used.

## 6.3 k-means clustering

K-means is one of the most popular and well-known clustering algorithms. In this method, first, k samples are considered as initial cluster centers. Then, distances between the points and these centers are calculated and the nearest points to each center are assigned to that cluster. Next, the mean of each cluster will be used as a new center. This process continues until that no changes appear in the clusters (Theodoridis & Koutroumbos, 2003). The k-means algorithm seeks to partition the data into k groups or clusters so that the within-group sum of squares is minimized (Webb, 2002).

## 6.4 Self-organizing map (SOM)

Self-organizing map or SOM is a kind of artificial neural network (ANN). It can be used for unsupervised clustering. This method was introduced by Kohonen in 1980 and their main application is to reduce the dimensional (Kohonen, 1998). In this method, topological structure of the input space will be saved. The net of neurons can be right angle or hexagonal grid and the adjacent cells upgrade during successive stages (Engelbrecht, 2002).

## 6.5 Cluster validity

Optimum number of clusters was found by silhouette method. In this method, a graphical validation was used for evaluating the number of clusters and comparing different scenarios. Therefore, by calculating the distances between samples in the clusters and distances between the prototypes the optimal number was determined (Rousseeuw, 1987).



## 7. Results and discussion

The aim of this study is to investigate the geochemical distribution of REEs. Therefore, the concentrations of REEs (after normalization between 0 and 1) were used as input data for clustering. But, after data processing, the clustering results were compared with concentrations of phosphorus and iron. Moreover, the lithology of samples were considered for validation.

Clustering results of four methods including modified basic sequential algorithmic scheme (MBSAS), hierarchical (agglomerative), k-means and SOM will be discussed in the following.

The input of the methods is a dataset of 42 vectors with 14 dimensions (42 samples and 14 rare earth elements). First, outliers should be put aside. Figure 3 shows the dendrogram based on average of each cluster and Euclidean distance between the clusters. Linkage analysis shows that two samples have more distance from others and can be put aside as

outliers. They are phosphorous iron ore with high concentrations of REEs. Contents of REEs in these two samples are much more in comparison to others. They were belonged to the certain clusters (due to the similarity) at the end of calculations.

**Figure 3 Linkage analysis with dendrogram based on average of each cluster and Euclidean distance between the clusters. Results show that two samples have more distance from others and can be put aside as outliers. They are in cluster 7 and 24 that are shown in the dendrogram.**

In MBSAS and hierarchical methods, two parameters (i.e. optimum threshold and number of clusters) should be identified. In this regard, the dendrogram were drawn. Figure 4 shows the dendrogram for identifying the optimum threshold and number of clusters. It has been calculated based on average of each cluster and Euclidean distance between the clusters. Optimum threshold was identified 0.4 based on dendrogram (Figure 4). In this way, 4 clusters are obtained. However, for all four methods, the number of clusters was changed in the range of 2-6, and then, results evaluated using silhouette criterion.

Finally, 4 clusters were decided as the optimal number. In this case the best results of silhouette values were attained for all methods. Silhouette plots for each method shows the validity of each sample in a certain cluster. Positive values shows that the sample has been clustered in the correct group and its magnitude is a measure of accuracy. Results of silhouette are shown in Figure 5. As it can be seen in Figure 5, one sample in MBSAS and hierarchical methods, has negative value. It means that this sample is in wrong cluster. Comparing the results of the methods shows that MBSAS and hierarchical had

the same outputs and so, the methods k-means and SOM have similar outputs. Moreover, results of k-means and SOM have the best matches with reality and experimental studies of samples and also field surveys.

**Figure 4 Dendrogram for identifying the optimum threshold and number of clusters.**

**Figure 5 Silhouette plots for each method shows the validity of each sample in a certain cluster. Positive values shows that the sample has been clustered in the correct group and its magnitude is a measure of accuracy.**

Characteristics of each cluster in each method are summarized in Table 2. For this purpose, averages of ∑REEs (total concentrations of rare earth elements) as well as P and Fe for each cluster have been calculated. Comparing these results with laboratory analyzes and field studies, concluded that samples can be classified in four types (Figure 8): (1) high anomaly (phosphorus iron ore), (2) low anomaly (metasomatized tuffs), (3) low anomaly (iron ore), and background (iron



ore and others). Since only the rare earth elements are used in this division, a good agreement of the results with lithology is considerable. Type 1 is comprised of iron ore with high anomaly of REEs (about 1900 ppm) and the high content of phosphorus (more than 2 %). Figure 2 shows SEM images of a sample from type 1. This type is the most prone for rare earth elements and containing apatite and monazite. However, fluorapatite is the main mineral of REEs in this type (due to the

XRD and SEM analyzes). The second type (i.e. metasomatized tuffs) has low anomaly of REEs, whereas the concentration of P is low. Samples of this group are metasomatized tuffs of andesite composition and mainly consist of actinolite and plagioclase with low concentrations of Fe and P, but the contents of REEs are considerable (with average about 400 ppm). SEM analysis shows that monazite is the mineral of REEs and apatite doesn't exist in this type. Third type shows low anomaly of REEs with lithology of ore body and relatively high contents of P (about 3400 ppm in SOM and k-means results

and about 1 % in MBSAS and hierarchical). The latter type is background (low concentrations of REEs) and comprised of various samples of iron ore and others (mainly metasomatic samples).

**Table 2 Characteristics of each cluster in each method. Iron and phosphorus concentrations are shown for comparison.**

As mentioned above, the results of k-means and SOM have the best matches with reality and experimental studies of samples and also field surveys. However, self-organizing map has the capability to present a two dimensional map (for

visual evaluation of clusters) from a multidimensional data. In addition, the weight distance matrix provides a tool to compare clusters. These advantages of SOM, make it more applicable for data processing in exploration works. Figure 6 (a) shows the topology of self-organizing map which has been used in this study as well as the number of samples for each cluster. Since Self- organizing map has a 2- dimensional topology, the relations between centers of 14- dimensional clusters have been illustrated in a 2- dimensional map. Weight distance matrix or unified distance matrix (U-matrix) is one of the

tools of SOM. Figure 6 (b) shows neighbor weight distances. Lines are used to display the relationship between neighbor neurons.  The darker the color, the further the distance between the neurons, as well as the lighter the color, the lesser the distance between the neurons. Therefore, as can be seen in Figure 6 (b), type 1 (i.e. high anomaly or phosphorus iron ore) has the maximum distance with type 3 and to a lesser degree with type 2. Also, type 3 and 4 are closest together and most similar to each other in terms of REEs. Finally, type 1 or phosphorus iron ore type is the most promising type for rare earth

elements. This type occurs mainly in supplementary part of pit 2 (or Pit 4).

**Figure 6 SOM topology and determining the number of samples for each cluster (a), SOM neighbor weight distances and neighbor connections (b)**

For a better comparison of the four methods, the outputs of clustering algorithms (Table 2) were normalized and the results were summarized in four bar chart (Figure 7).

**Figure 7 comparative bar charts of normalized values of REE, P and Fe for all clustering methods.**

In this study, pattern recognition helped to divide the samples in appropriate groups, according to the contents of REEs and results are consistent with the concentration of P and also lithology of the samples. Variety of parameters, especially in case of REEs explorations makes some complication for interpretation of data and exploration area. Since, single-variable methods don't provide useful information, the authors proposed four common clustering algorithms which have been



explained above. The output of these four methods (Figure 7 and Table 2) shows that the discrimination of clusters is based on the lithology of the samples, in addition to the REEs. Therefore, it is proven that proposed methods have found the relation between the distribution of REEs and lithology of study area. In this regards, we claim that pattern recognition helps to find some hidden informations associated with complicated nature of rare earth elements systems. Figure 8 is prepared to

show the application and efficiency of unsupervised methods in evaluating geochemical distribution of rare earth elements (REEs) in the Kiruna type magnetite– apatite deposit of Se-Chahun, while it doesn't need to do additional geological studies and extra cost and time.

**Figure 8 some examples for samples which have been classified in four types along with their microscopic images. (a) type 1: high anomaly (phosphorus iron ore), sample 4-1, iron ore sample including apatite and monazite; (b) type 2: low anomaly**
**(metasomatized tuffs), sample 2-6, including actinolite, calcite, feldspar and monazite; (c) type 3, low anomaly (iron ore), sample 4-6, iron ore sample including apatite and monazite; (d) type 4: background (iron ore and others), sample 1-16, metasomatite including plagioclase, feldspar and actinolite. Abbreviations: Ap= apatite, Mnz= monazite, Act= actinolite, Mag= magnetite and Pl= plagioclase.**

Four methods including modified basic sequential algorithmic scheme (MBSAS), hierarchical (agglomerative), k-means
and SOM were applied in this study. However, k-means and SOM are more advanced in comparison to others. They improve and modify the weights or centers of the clusters, continuously in several stages. In contrast, MBSAS and hierarchical are more simple and elementary. Because, the centers of clusters are determined in one stage. Furthermore, SOM has an advantage that the distances between the clusters can be assessed visually in a two dimensional map (Figure 6, b). Since the input dataset is comprised of 14 dimensional vectors (14 REEs), SOM is a good tool for evaluating it in a two dimensional
space.

## 8. Conclusion

Following points were concluded:

- Successful clustering of dataset which is consistent with geological facts, laboratory and field studies was achieved.
- The results of k-means and SOM have the best matches with reality and experimental studies of samples and also
field surveys.
- Since only the rare earth elements are used in this division, a good agreement of the results with lithology is considerable.
- Results showed that the unsupervised pattern recognition helps to find some hidden informations which would be difficult to achieve in usual ways (i.e. finding the appropriate clusters). Methods which have been presented in this
study will help better interpretation of data, despite there are many variables.
- It concluded that a combination of numerical models and geological studies leads to have the best outputs and outcomes in exploration programs of REEs.
- Proposed methods help to reduce the time and cost by eliminating the additional geological studies.





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

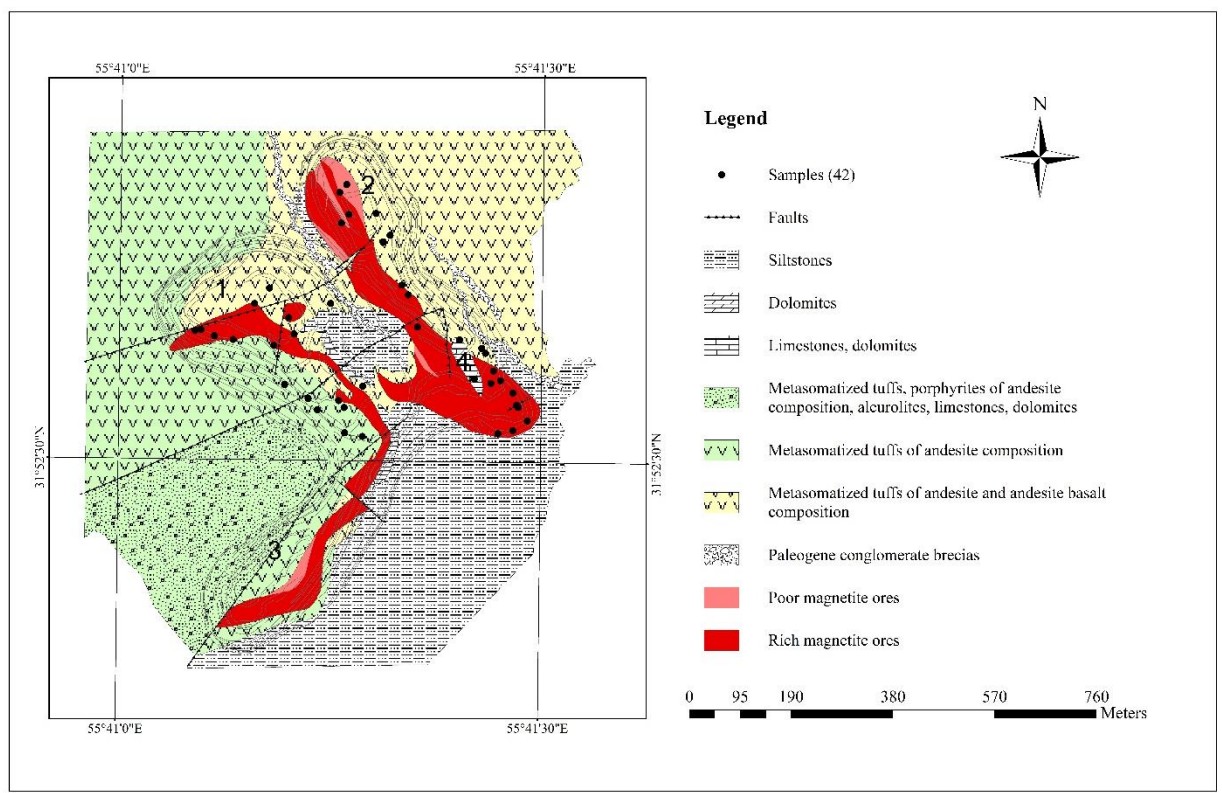

**Figure 1 Geological map of Se-Chahun deposit (anomaly X) and sample locations. Contours of open pits are shown in the map and the open pits are numbered from 1 to 4 (supplementary part of pit 2 are known as pit 4) (modified after NISCO, 1975).**





**Figure 2 Back-scattered electron images of a sample from phosphate rocks. Abbreviations: Ap= apatite, Mnz= monazite and Mag= magnetite.**





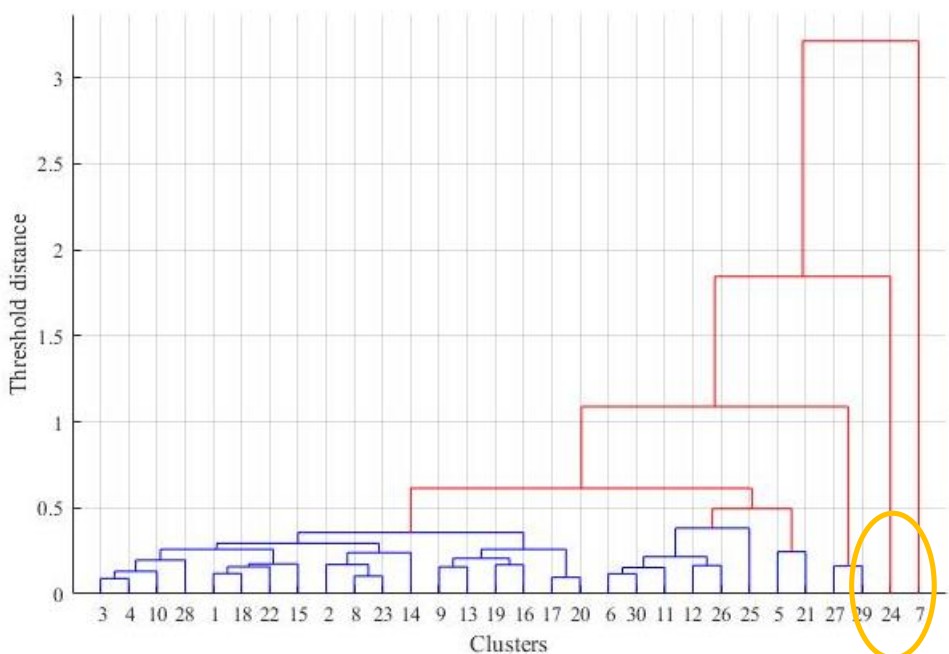

**Figure 3 Linkage analysis with dendrogram based on average of each cluster and Euclidean distance between the clusters. Results show that two samples have more distance from others and can be put aside as outliers. They are in cluster 7 and 24 that are shown in the dendrogram.**

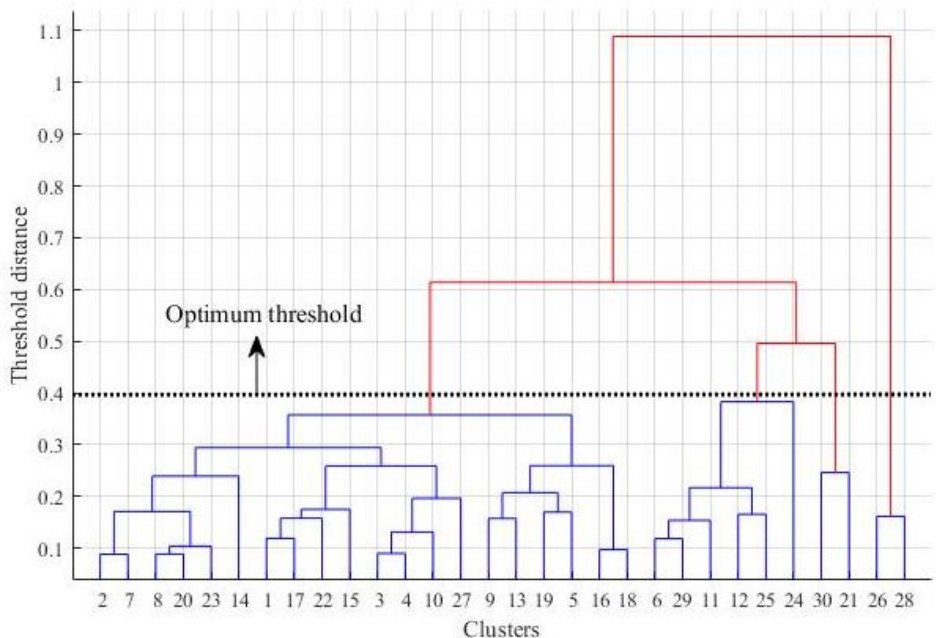

**Figure 4 Dendrogram for identifying the optimum threshold and number of clusters.**





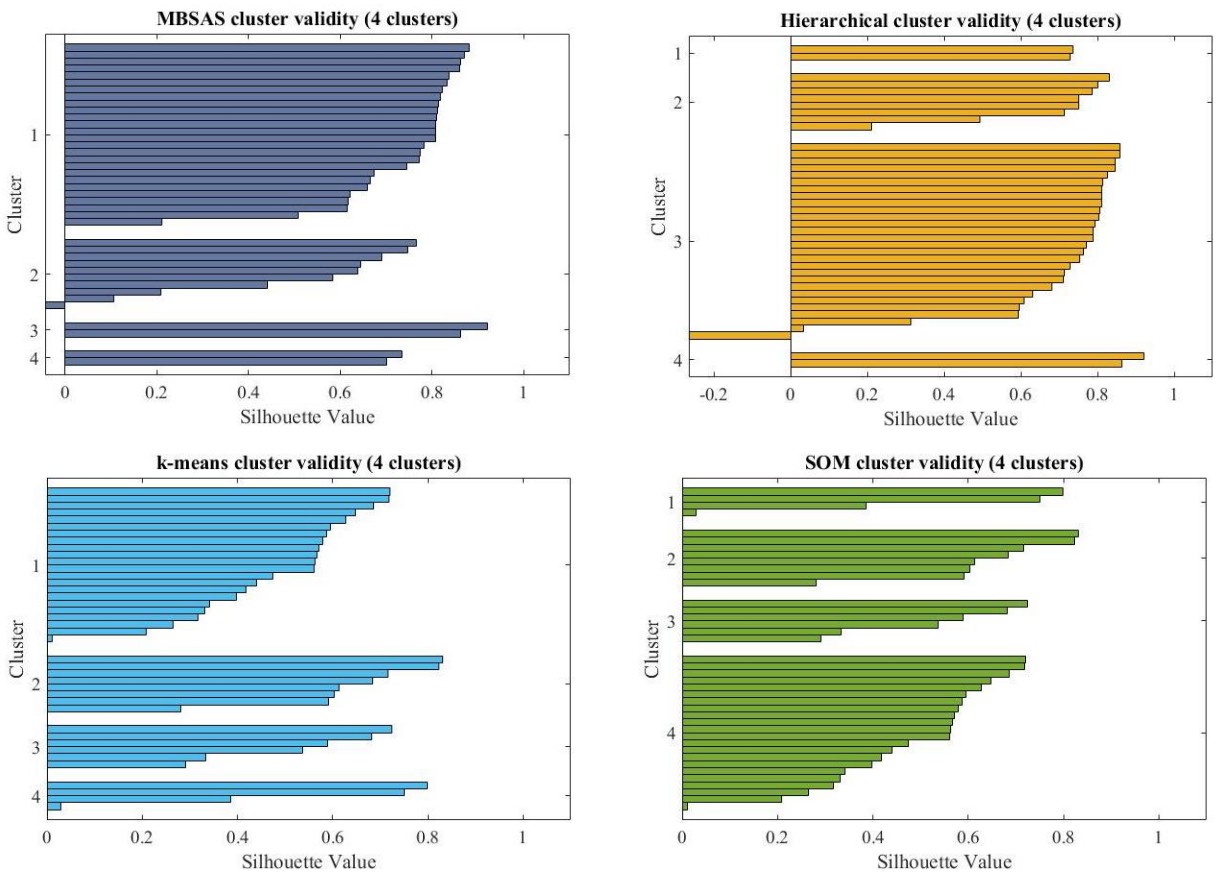

**Figure 5** Silhouette plots for each method shows the validity of each sample in a certain cluster. Positive values shows that the sample has been clustered in the correct group and its magnitude is a measure of accuracy.

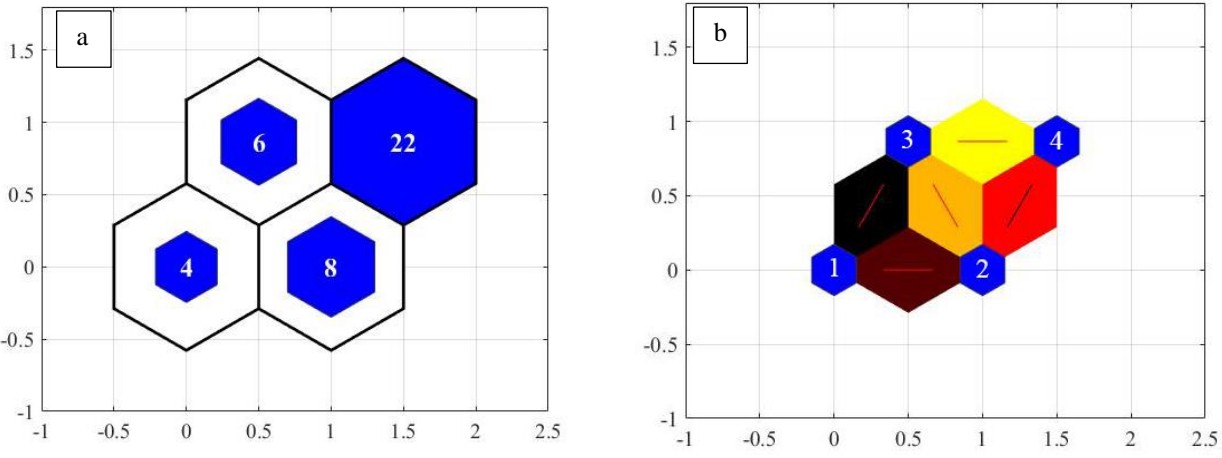

**Figure 6 SOM topology and determining the number of samples for each cluster (a), SOM neighbor weight distances and neighbor connections (b)**



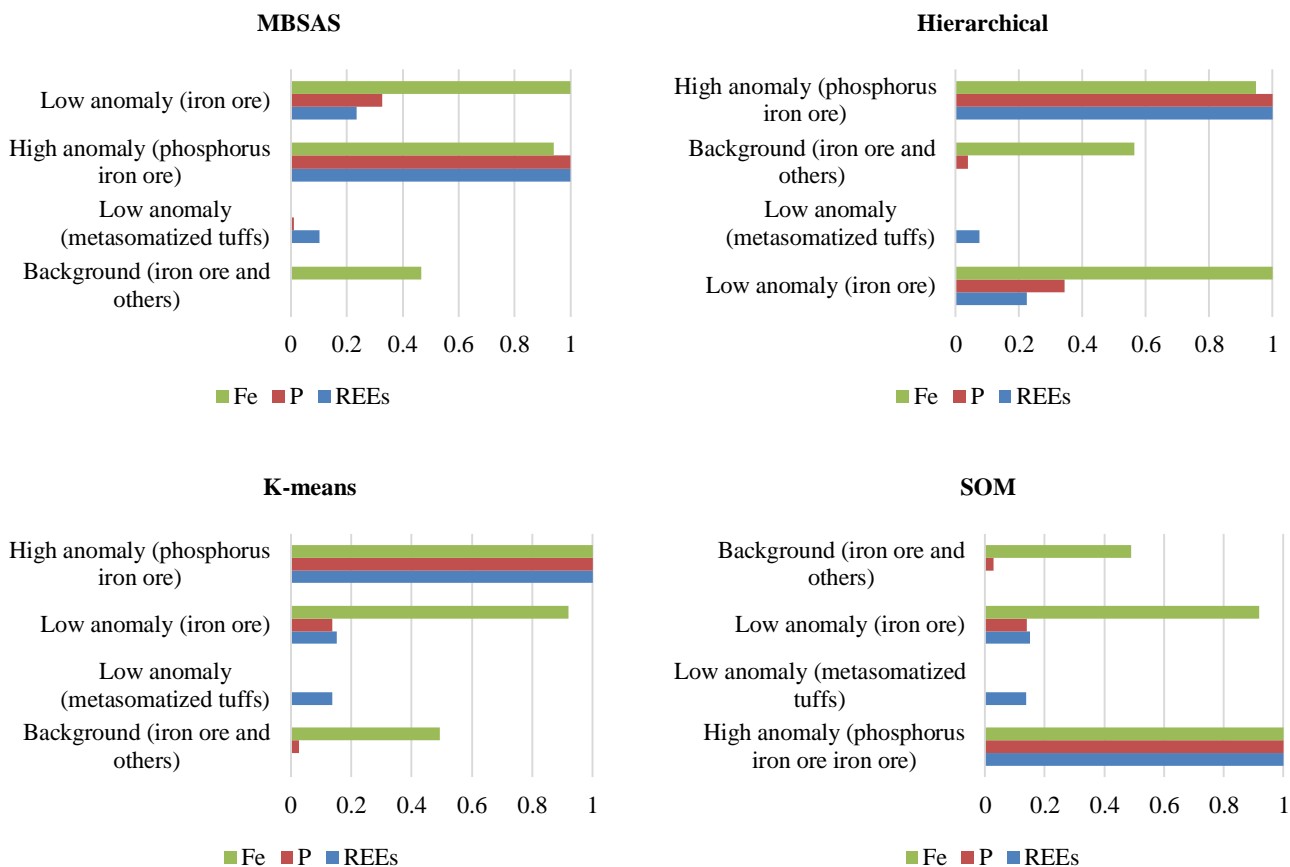

**Figure 7 comparative bar charts of normalized values of REE, P and Fe for all clustering methods.**



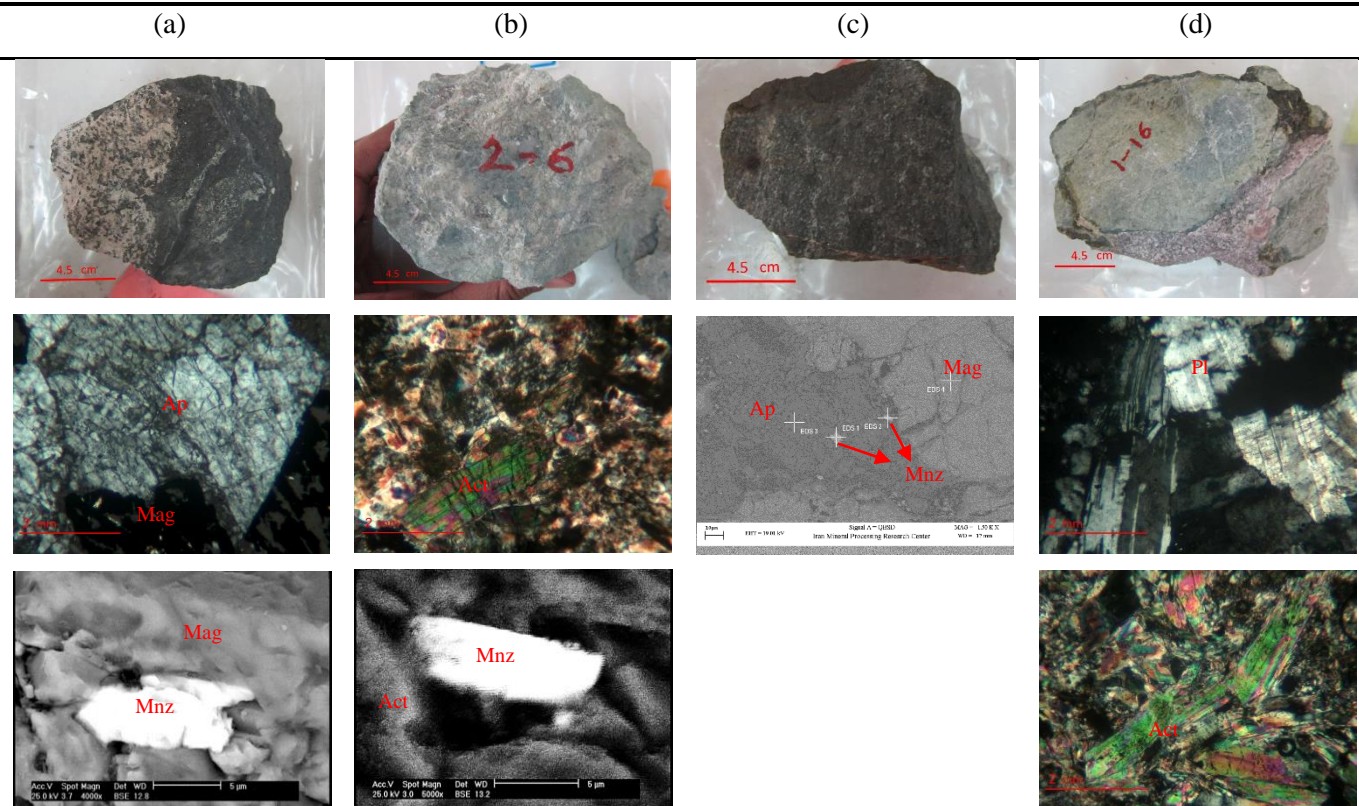

**Figure 8 some examples for samples which have been classified in four types along with their microscopic images. (a) type 1: high anomaly (phosphorus iron ore), sample 4-1, iron ore sample including apatite and monazite; (b) type 2: low anomaly (metasomatized tuffs), sample 2-6, including actinolite, calcite, feldspar and monazite; (c) type 3, low anomaly (iron ore), sample 4-6, iron ore sample including apatite and monazite; (d) type 4: background (iron ore and others), sample 1-16, metasomatite including plagioclase, feldspar and actinolite. Abbreviations: Ap= apatite, Mnz= monazite, Act= actinolite, Mag= magnetite and Pl= plagioclase.**

Table 1 mean, variance, minimum and maximum of 14 assayed rare earths in 42 samples.

| Elements (ppm) | La | Ce | Pr | Nd | Sm | Eu | Gd | Tb | Dy | Er | Tm | Yb | Lu | Y |
|---|---|---|---|---|---|---|---|---|---|---|---|---|---|---|
| Mean | 73 | 154 | 20 | 75 | 13 | 2 | 13 | 2 | 12 | 7 | 1 | 13 | 1 | 56 |
| Variance | 27180 | 111800 | 1202 | 15180 | 299 | 3 | 242 | 4 | 121 | 40 | 1 | 64 | 0 | 2869 |
| Minimum | 3 | 2 | 0 | 5 | 1 | 0 | 1 | 0 | 1 | 1 | 0 | 1 | 0 | 9 |
| Maximum | 995 | 2037 | 203 | 740 | 102 | 9 | 90 | 12 | 60 | 32 | 4 | 42 | 3 | 305 |

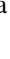
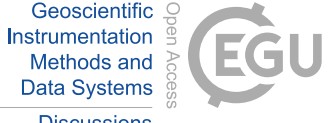

**Table 2 Characteristics of each cluster in each method. Iron and phosphorus concentrations are shown for comparison.**

**MBSAS**

| Cluster No. | ∑REEs (ppm) | P (ppm) | Fe% | Description |
|---|---|---|---|---|
| 1 | 169 | 933 | 29 | Background (iron ore and others) |
| 2 | 422 | 1294 | 10 | Low anomaly (metasomatized tuffs) |
| 3 | 2646 | 31934 | 47 | High anomaly (phosphorus iron ore) |
| 4 | 749.14 | 11061 | 49 | Low anomaly (iron ore) |

**Hierarchical**

| Cluster No. | ∑REEs (ppm) | P (ppm) | Fe% | Description |
|---|---|---|---|---|
| 1 | 749 | 11061 | 49 | Low anomaly (iron ore) |
| 2 | 383 | 123 | 4 | Low anomaly (metasomatized tuffs) |
| 3 | 199 | 1335 | 29 | Background (iron ore and others) |
| 4 | 2646 | 31934 | 47 | High anomaly (phosphorus iron ore) |

**k-means**

| Cluster No. | ∑REEs (ppm) | P (ppm) | Fe% | Description |
|---|---|---|---|---|
| 1 | 143 | 775 | 25 | Background (iron ore and others) |
| 2 | 383 | 143 | 3 | Low anomaly (metasomatized tuffs) |
| 3 | 407 | 3387 | 44 | Low anomaly (iron ore) |
| 4 | 1887 | 23585 | 48 | High anomaly (phosphorus iron ore) |

**SOM**

| Cluster No. | ∑REEs (ppm) | P (ppm) | Fe% | Description |
|---|---|---|---|---|
| 1 | 1887 | 23585 | 48 | High anomaly (phosphorus iron ore iron ore) |
| 2 | 383 | 123 | 4 | Low anomaly (metasomatized tuffs) |
| 3 | 407 | 3387 | 44 | Low anomaly (iron ore) |
| 4 | 143 | 775 | 25 | Background (iron ore and others) |