# Peer review of "Application of unsupervised pattern recognition approaches for exploration of rare earth elements in Se-Chahun iron ore, central Iran."

_Geoscientific Instrumentation, Methods and Data Systems, 2017_

## Short Comment (SC1) · 12 Sep 2017

This is Dear authors I wish to congratulate you for writing this really helpful paper. I am interested in the application of numerical modeling methods in geosciences. I enjoy the novelty of this paper that applies clustering approaches for the study of rare earth elements (REEs). Due to the diversity of variables in the studies of REEs, pattern recognition techniques are very useful to unravel subtleties in their distribution. I highly

recommend this paper and in fact, this paper could stand as a seminal reference in the geochemical studies of REEs in Iran. There are few papers dealing with the geochemical studies of REEs in Iran. However, there is a considerable economic potential of REEs in Central Iran and therefore future studies should be focused in this field of study. Best regards, Mohammad Parsa,

---

## Author Comment (AC1) · 12 Sep 2017

Dear Mohammad Parsa

Thank you very much for the positive comments on our article.

Kind regards, Mohammadali Sarparandeh

---

## Referee Comment (RC1) · A. Mollajan (Referee) · 20 Sep 2017

Dear author The paper has provided valuable results and information that have made this research remarkable. Practicality and efficiency of the discussed methods with combination of the geological studies are the strengths of this article. In addition, a good discrimination based on lithology which was attained just by using REEs is noticeable. The English language is well-understandable, and is written clearly, so has

the merit of a scientific publication. However, I have some comments as follows: 1- It is better to define the abbreviations: SEM, XRD, and ICP-MS in abstract, before using them. 2- The word "and" at the end of line 19 should be omitted. 3- I think that figure 3 is not necessary. 4- There are good examples of the samples in figure 8. I suggest to add some explanation about them in the section 3 (mineralogy and lithology). Best regards

---

## Author Comment (AC2) · 20 Sep 2017

Dear Dr. Mollajan

Thank you for your positive evaluation. Your comments helped to improve the quality of the article. We followed your advices and did the corrections in final version of the manuscript. Therefore:

[Figure]

1- We defined all the abbreviations in the first use of them. 2- The sentence was corrected (page 4, line 19). 3- We removed figure 3 and just explained it in the text. 4- The title of section 3 was changed to "mineralogy and lithology" and we explained about the samples.

Best Regards, Mohammadali Sarparandeh

---

## Referee Comment (RC2) · Anonymous Referee #1 · 26 Oct 2017

The aim of this study is to investigate the geochemical distribution of REEs . for this purpose the paper introduces the use of efficient methods for data processing in the field of earth science, in particular for high-dimensional data such as geochemical data. So the main issue of this paper is application of unsupervised pattern recognition approaches in evaluating geochemical distribution of rare earth elements (REEs) in the Kiruna type magnetite– apatite deposit of Se-Chahun. However, the manuscript do not provide substantial new information of interest. I do believe that the manuscript needs

additional theoretical development and explication of the data and analyses before it could be sent to publish. As written, the paper does not provide sufficient theoretical motivation and does not provide enough information about state of art about the four methods (modified basic sequential algorithmic scheme (MBSAS), hierarchical (agglomerative), k-means and SOM) applied in this field of study. Further, the data and analyses are not sufficiently justified.

———————————————

---

## Author Comment (AC3) · 28 Oct 2017

Dear Anonymous Referee #1

We really appreciate you for very useful scientific comments. They helped us to promote the quality of the paper. We carefully considered all comments and made the relevant changes. Some additional explanations and Figure 7 had been added to initial version of the manuscript (after quick review). They are highlighted by yellow. In

addition, new changes were applied in the marked-up manuscript version (using track changes in Word). The comments and author's responses and changes were classified as following: ïČij

comments from Reviewers

1 The manuscript does not provide substantial new information of interest. The paper does not provide sufiňĄcient theoretical motivation and does not provide enough information about state of art about the four methods (modified basic sequential algorithmic scheme (MBSAS), hierarchical (agglomerative), k-means and SOM) applied in this field of study.

2 The manuscript needs additional theoretical development and explication of the data and analyses. The data and analyses are not sufficiently justified. ïČij

author's response

1- Potential of Central Iran for REEs needs more attention. Exploration aspects of REEs was studied by authors in Se-Chahun deposit using bulk lithology samples for first time. Moreover, the application of unsupervised pattern recognition in exploration of REEs has not been considered before. We have used new original dataset which has been produced by ourselves. We mentioned in the text: "In this study, pattern recognition helped to divide the samples in appropriate groups, according to the contents of REEs and results are consistent with the concentration of P and also lithology of the samples. Variety of parameters, especially in case of REEs explorations makes some complication for interpretation of data and exploration area. Since, single-variable methods don't provide useful information, the authors proposed four common clustering algorithms". We think that this paper suggests a new approach for exploration of REEs which is more applicable and compatible with multi-variate nature of them. As it was mentioned in line 15 of the first page, the proposed method helps to find some hidden information from dataset which are not easily achievable in usual. For example, in this study we found a cluster that had not been considered before. It

was previously believed that the concentration of rare elements is directly related to apatite and consequently phosphorus. This hypothesis is generally correct. However, in cases with medium amounts of REEs, we found that there is a different condition. In fact, we have another group of samples in which there are lesser amounts of P with considerable concentrations of REEs. This group of data was separated easily by clustering methods. It confirmed by evaluation of samples under SEM. After a complete survey of samples under SEM, we found that the samples of this cluster (figure 7, b) contain monazite but no apatite. Since only the rare earth elements were used in this division, a good agreement of the results with lithology is considerable (line 21).

2- Some explanations were added to complete and explication of the data and analyses (lines 25-28 of page 3, lines 11-20 of page 8 and Table 3). Representative samples of Figure 7 and Table 3 and mentioned explanations explicate the data and analyses and show the performances of clustering methods.

author's changes in manuscript

All the changes are marked in "marked-up manuscript version". The changes after quick review were highlighted by yellow color. The changes which are related to Referee #2 were highlighted by green. Recently changes were marked using track changes in Word.

Best regards,

Authors.

Please also note the supplement to this comment:
https://www.geosci-instrum-method-data-syst-discuss.net/gi-2017-38/gi-2017-38-AC3-supplement.zip